# ON THE EFFICACY OF GROUP-WISE CLIPPING IN DIFFERENTIALLY PRIVATE OPTIMIZATION

## ABSTRACT

Recent advances have substantially improved the accuracy, memory cost, and training speed of differentially private (DP) deep learning, especially on large vision and language models with millions to billions of parameters. In this work, we thoroughly study the per-sample gradient clipping style, a key component in DP optimization. We show that different clipping styles have the same time complexity but instantiate an accuracy-memory trade-off: while the all-layer clipping (of coarse granularity) is the most prevalent and usually gives the best accuracy, it incurs heavier memory cost compared to other group-wise clipping, such as the layer-wise clipping (of finer granularity). We formalize this trade-off through our convergence theory and complexity analysis. Importantly, we demonstrate that the accuracy gap between group-wise clipping and all-layer clipping becomes smaller for larger models, while the memory advantage of the group-wise clipping remains. Consequently, the group-wise clipping allows DP optimization of large models to achieve high accuracy and low peak memory simultaneously.

## 1 INTRODUCTION

Differentially private (DP) optimization of deep learning models has enjoyed amazing accuracy and rigorous guarantee against privacy risks. For example, recent successes of DP GPT2 Li et al. (2021); Bu et al. (2022b); Yu et al. (2021a) have achieved 64.6 BLEU score (considered as 'often better than human') at strong privacy guarantee ($\epsilon = 3$), on the E2E restaurant review dataset. This is only marginally below the standard non-private GPT2 which achieves 66.8 BLEU. On computer vision tasks, under strong privacy guarantee $\epsilon = 2$, DP vision models have achieved $97.1\%/86.2\%$ accuracy on CIFAR10/100 by Bu et al. (2022a) and over $81\%$ accuracy on ImageNet by De et al. (2022); Mehta et al. (2022).

These advances are realized through DP optimization, which applies the standard SGD/Adam on the *private gradient* (1) instead of the regular gradient $\sum_i \boldsymbol{g}_i$:

$$\text{DP-SGD:} \quad \mathbf{w}_{t+1} = \mathbf{w}_t - \eta_t \mathbf{G}_{\text{private}}, \text{ where } \mathbf{G}_{\text{private}} := \sum_i \boldsymbol{g}_i \cdot C(\boldsymbol{g}_i; R) + \sigma_{\text{DP}} R \cdot \mathcal{N}(0, \mathbf{I}). \quad (1)$$

Here $\boldsymbol{g}_i$ is the per-sample gradient of loss $L_i$ to the model parameter $\mathbf{w}$, $\eta_t$ is the learning rate, $\sigma_{\text{DP}}$ is the noise level to account for the privacy loss Abadi et al. (2016); Mironov (2017); Dong et al. (2019); Bu et al. (2020); Gopi et al. (2021); Zhu et al. (2021); Koskela et al. (2020), and $C$ is the clipping factor from some clipping function, so that $\boldsymbol{g}_i \cdot C(\boldsymbol{g}_i)$ performs the per-sample gradient clipping. For instance, one can use the Abadi's (Abadi et al., 2016) or the automatic (AUTO) clipping function (Bu et al., 2022b), both giving an equally strong privacy guarantee. One important yet under-studied subject in DP optimization is the per-sample gradient clipping style. In most of the existing works, the privatization (1) takes place on the gradient of all trainable parameters, known as the all-layer or flat clipping style Abadi et al. (2016). In fact, this widely-used style can be generalized to the group-wise gradient clipping McMahan et al. (2018a), which can improve the memory efficiency at the cost of possibly worse accuracy. Generally speaking, the group-wise clipping assigns the trainable parameters to $M$ groups and privatizes each group separately. We denote the parameters in the $m$-th group as $\mathbf{W}^{(m)}$ and the corresponding per-sample gradient $\boldsymbol{g}_i^{(m)} = \partial L_i / \partial \mathbf{W}^{(m)}$, whereas the whole gradient is concatenated as $\boldsymbol{g}_i = [\boldsymbol{g}_i^{(1)}, \boldsymbol{g}_i^{(2)}, \cdots, \boldsymbol{g}_i^{(M)}]$. A vector of clipping thresholds $[R_1, R_2, \cdots, R_M] \in \mathbb{R}^M$ are applied to each group, so that the $m$-th group's private gradient is

$$\mathbf{G}_{\text{private}}^{(m)} = \sum_i \boldsymbol{g}_i^{(m)} C(\boldsymbol{g}_i^{(m)}; R_m) + \sigma_{\text{DP}} \|[R_1, R_2, \cdots]\| \cdot \mathcal{N}(0, \mathbf{I}_m).$$

In this context, the all-layer clipping means $M = 1$ and another example is the layer-wise or per-layer clipping McMahan et al. (2018b); Bu et al. (2021); He et al. (2022), which treats the parameters (weights and biases) in each layer as one group, hence $M$ equals the number of layers in the neural network. As we will show, the choice of group-wise clipping style has significant influences over the convergence and thus the accuracy (see Figure 1), the computational efficiency, and the algorithmic design (see Figure 2).

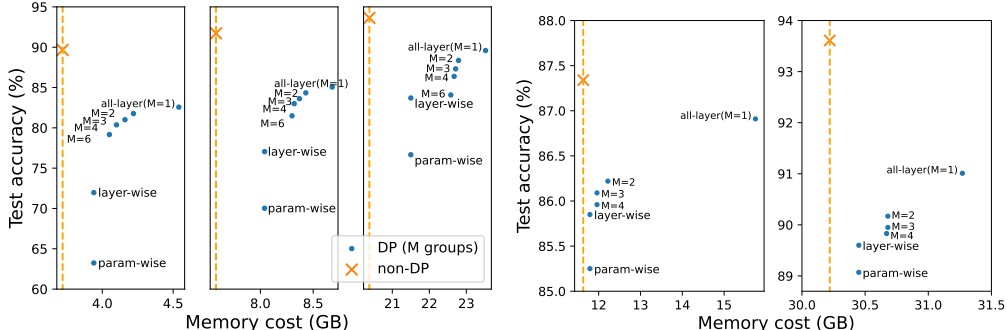

Figure 1: Accuracy and memory on CIFAR100 ($\epsilon = 2$, virtual batch size 50, left to right: ViT-small/base/large) and QNLI ($\epsilon = 3$, virtual batch size 40, left to right: RoBERTa-base/large).

Orthogonal to the per-sample gradient clipping style, a long list of researches have been devoted to efficiently implement the DP optimization, by increasing the training speed and/or reducing the memory cost. One approach is to improve the computational efficiency algorithmically, without affecting the accuracy. For instance, the slowdown of DP optimization (compared to non-DP optimization) has been improved from $24\times$ on small CNN with Tensorflow XLA compiler Subramani et al. (2021), to $9\times$ in JAX De et al. (2022), to roughly $2\times$ on GPT2 / RoBERTa / ViT by the ghost clipping technique Li et al. (2021); Bu et al. (2022a), and finally to $1.1\times$ by the Book-Keeping (BK) technique Bu et al. (2022c). In this work, the clipping styles are implemented by the BK algorithm for the best algorithmic efficiency.

In this work, we study the group-wise clipping style in depth, with a specific emphasis on its convergence and its algorithmic relation to the back-propagation. We observe that the family of group-wise clipping instantiates an accuracy-memory trade-off (i.e. more groups, better memory, worse accuracy), whose two endpoints are the all-layer and layer-wise clipping. In fact, we can group the trainable parameters so as to achieve the best DP accuracy and low memory cost simultaneously (to be demonstrated in Figure 6).

## 1.1 CONTRIBUTIONS

1. **[Novel clipping styles]** We propose novel choices of the group-wise clipping that are equally fast and private. The uniform clipping is easy to design and instantiates an accuracy-memory trade-off to select; the non-uniform clipping can achieve high accuracy and low memory cost beyond the trade-off, though being harder to design.

2. **[A convergence theory]** We provide the first convergence result of the group-wise clipping in Theorem 1, showing that DP-SGD has the same asymptotic convergence rate $O(T^{-1/4})$ as the standard SGD, but the convergence guarantee worsens as the number of groups increases.

3. **[Guaranteed algorithmic efficiency]** We implement our group-wise clipping efficiently so that all group-wise clipping enjoy almost the same training speed as the standard non-DP optimization. This contrasts with prior work which claims that all-layer clipping is about $1.5 \sim 2\times$ slower than layer-wise clipping (see (He et al., 2022, Figure 1b)).

4. **[Peak memory profile]** We provide an explicit memory profile in Fact 6.1, which explains the peak memory of DP optimization and guides the design of group-wise clipping towards larger batch size and faster training.

5. **[New baselines]** We experiment different choices of group-wise clipping on a range of new DP tasks. We empirically demonstrate that, with the proper group-wise clipping, DP optimization can achieve better accuracy at lower memory cost.

## 1.2 RELATED WORKS

Group-wise clipping can be implemented via different algorithms in Tensorflow-privacy, Opacus Yousefpour et al. (2021), FastGradClip Lee & Kifer (2020), private-transformer Li et al. (2021)), private-vision Bu et al. (2022a), FastDP (using BK algorithm Bu et al. (2022c)). We use BK to implement the group-wise clipping style in Algorithm 1, due to its state-of-the-art efficiency on large-scale vision and language tasks. Notice that, prior to this work, BK algorithms only comes with the all-layer clipping style.

The general concept of group-wise clipping covers a family of gradient clipping styles. The most popular one is the all-layer clipping, which groups all layers into one group and usually enjoys the highest accuracy among other clipping styles. The layer-wise clipping instead groups each layer into a group, thus requiring a long vector of clipping thresholds when the model is of hundreds of layers. These additional hyperparameters $[R_1, R_2, ...]$ are difficult to tune manually and oftentimes introduce extra privacy risk if tuned adaptively to the data Andrew et al. (2021); He et al. (2022). Similarly, the parameter-wise clipping used by Opacus Yousefpour et al. (2021) groups each parameter (weight and bias) into a group[1]. McMahan et al. (2018a) proposes the type-wise clipping such that linear layers form a group and convolution layers form another group. In distributed learning, large models are partitioned into multiple devices, each of which defines a group according to the per-device clipping He et al. (2022).

In contrast to existing works which focus on specific clipping styles, we explore the whole class of group-wise clipping, thus to reveal an accuracy-memory trade-off. Different from the empirical nature in the literature, we give the first convergence theory of group-wise clipping, and the first peak memory profile from the complexity analysis. We emphasize that the choice of group-wise clipping style can serve as a strong alternative to the adaptive clipping threshold Andrew et al. (2021); He et al. (2022).

Table 1: List of group-wise per-sample gradient clipping styles.

| Clipping style | Reference |
|---|---|
| all-layer | Abadi et al. (2016) |
| layer-wise | McMahan et al. (2018b) |
| param-wise | Yousefpour et al. (2021) |
| type-wise | McMahan et al. (2018a) |
| per-device | He et al. (2022) |
| uniform | this work |
| non-uniform | this work |

## 2 PRELIMINARIES

### 2.1 DIFFERENTIAL PRIVACY IN DEEP LEARNING

We work with the $(\epsilon, \delta)$-DP by Dwork et al. (2006), where strong DP is indicated by small $(\epsilon, \delta)$ and means it is difficult for any privacy attacker to distinguish or detect an arbitrary training sample.

**Definition 2.1** (Dwork et al. (2006)). A randomized algorithm $M$ is $(\varepsilon, \delta)$-DP if, for any two neighboring datasets $S, S'$ that differ by one data point and for any event $E$,

$$\mathbb{P}[M(S) \in E] \leqslant e^{\varepsilon} \mathbb{P}[M(S') \in E] + \delta. \tag{2}$$

In deep learning, DP is realized by applying SGD, AdamKingma & Ba (2015), LAMBYou et al. (2019), FedAvgMcMahan et al. (2017), etc. on the private gradient (1) with respect to the trainable parameters, which are partitioned into $M$ groups and assigned with $M$ clipping thresholds.

We now declare our setting throughout this work:

1. We use automatic (AUTO) clipping function $C_i = C(\boldsymbol{g}_i) = \frac{1}{\|\boldsymbol{g}_i\|_2 + 0.01}$;
2. We use Renyi DP (Mironov, 2017) accoutant for all experiments;
3. We use non-adaptive clipping threshold $R_m = 1/\sqrt{M}$ (McMahan et al., 2018b);
4. We use the BK algorithm (Bu et al., 2022c) as our backbone DP implementation.

### 2.2 BACK-PROPAGATION

The efficiency of DP optimization is critically determined by that of the per-sample gradient clipping, which can be implemented with marginal overhead by the BK algorithm Bu et al. (2022c).

---

[1]Note that Opacus(v1.3) claims to support layer-wise clipping though they actually support parameter-wise clipping; see its CIFAR10 example Line 341.

Specifically, the BK algorithm makes DP optimization (with the all-layer clipping) almost as efficient as the non-DP optimization, by re-arranging the computation of output gradients and parameter gradients. To see this, we describe two sub-processes of the back-propagation: consider a linear layer (the $l$-th layer),[2] $\boldsymbol{a}_{(l+1)} = \phi(\boldsymbol{s}_{(l)}) = \phi(\boldsymbol{a}_{(l)}\mathbf{W}_{(l)})$, where $\boldsymbol{a} \in \mathbb{R}^{BTd}$ is layer's input (here $B$ being batch size, $T$ being sentence length or number of pixels), $\boldsymbol{s} \in \mathbb{R}^{BTp}$ is layer's output, $\mathbf{W} \in \mathbb{R}^{dp}$ is weight, and $\phi$ is any inter-layer operation like ReLU or pooling.

During the forward propagation, $\boldsymbol{a}_{(l)}$ is computed and stored. During the back-propagation, at each layer, the output gradient $\frac{\partial L}{\partial \boldsymbol{s}_{(l)}}$ is computed and then produces the parameter gradient:

$$\text{Standard: } \frac{\partial L}{\partial \mathbf{W}_{(l)}} = \frac{\partial \sum_i L_i}{\partial \mathbf{W}_{(l)}} = \boldsymbol{a}_{(l)}^\top \frac{\partial L}{\partial \boldsymbol{s}_{(l)}}, \text{ DP: } \frac{\partial \sum_i C_i L_i}{\partial \mathbf{W}_{(l)}} = \boldsymbol{a}_{(l)}^\top \text{diag}(C_1, \cdots, C_B) \frac{\partial L}{\partial \boldsymbol{s}_{(l)}}. \quad (3)$$

Here the per-sample gradient norm (or the clipping factor $C_i$) can be computed at small cost, i.e. $< 10\%$ memory overhead and $\approx 20\%$ slowdown for large models (see Figure 5 in Bu et al. (2022c)).

## 2.3 CLIPPING THRESHOLDS

Tuning the clipping threshold vector $\{R_m\} \in \mathbb{R}^M$ can be expensive for a network with hundreds of layers. The simplest choice is to use the same clipping threshold for all groups McMahan et al. (2018b): $R_1 = \cdots = R_M = R/\sqrt{M}$. Such a choice is data-independent and model-driven, and we adopt this for the AUTO private gradient: for the $m$-th group,

$$\mathbf{G}_{\text{private}}^{(m)} = \sum_i \frac{\boldsymbol{g}_i^{(m)}}{\sqrt{M}(\|\boldsymbol{g}_i^{(m)}\|_2 + 0.01)} + \sigma_{\text{DP}} \cdot \mathcal{N}(0, \mathbf{I}_m). \quad (4)$$

It is also possible to use the adaptive data-driven clipping threshold Andrew et al. (2021); He et al. (2022); Golatkar et al. (2022), although it needs either extra training data or extra privacy budget. For example, one can use 90% quantile of per-sample gradient norms from the public data as the clipping threshold on the private data, or use a second DP-SGD to learn the adaptive clipping thresholds as hyperparameters, thus adding to the computation cost and privacy budget. Empirically, the benefit of adaptive clipping threshold is insignificant, as illustrated in Table 2.

Table 2: Test accuracy of SST2 dataset at $\epsilon = 3$. Results other than ours are from He et al. (2022).

|  | Clipping style | 10 epochs | 20 epochs | 30 epochs |
|---|---|---|---|---|
| RoBERTa-base | all-layer | 90.53 | 90.76 | 91.27 |
|  | layer-wise (adaptive) | 91.30 | 91.57 | 92.10 |
|  | all-layer (ours) | 92.32 | 92.66 | 93.00 |
|  | layer-wise (ours) | 92.09 | 92.43 | 92.55 |
| RoBERTa-large | all-layer | 93.00 | 93.50 | 93.90 |
|  | layer-wise (adaptive) | 92.80 | 93.63 | 93.67 |
|  | all-layer (ours) | 94.50 | 94.84 | 94.95 |
|  | layer-wise (ours) | 94.15 | 94.27 | 94.38 |

## 3 ALGORITHM FOR GROUP-WISE CLIPPING

In this section, we modify the back-propagation so as to efficiently implement the group-wise clipping. We note that in (3), to derive the clipping factor $C_i$, the output gradients are book-kept until all layers in the current group have been back-propagated. This is visualized in Figure 2 as the stacking of different colors, which represent the computation of output gradients and parameter gradients: consider an 100-layer network and $M = 50$, then the group-wise clipping factor $C_i$ is computed only if two layers have been back-propagated; when $M = 1$ (all-layer clipping), $C_i$ is computed after all 100 layers have been back-propagated.



Figure 2: Back-propagation of BK algorithm with the group-wise clipping style.

---

[2]Convolution and embedding layers are equivalent to linear layers (see Li et al. (2021); Bu et al. (2022a)), which contain $\approx 99.9\%$ of model parameters (see Table 7 in Bu et al. (2022c)).

**Remark 3.1.** The layer-wise/param-wise clipping does not re-arrange the order of back-propagation (see the similarity of colors in Figure 2). This feature is particularly desirable for distributed learning, where back-propagation involves communication and rewriting the orchestration is hard.

In Algorithm 1, we implement our group-wise clipping in DP optimization, following Figure 2[3].

---

**Algorithm 1** Differentially private optimization with group-wise clipping

---

**Parameters:** $l$-th layer weights $\mathbf{W}_{(l)}$, $m$-th group weights $\mathbf{W}^{(m)}$, noise level $\sigma_{\mathrm{DP}}$.

1: **for** layer $l = 1, 2, \cdots$ **do**
2:     Get activation $\{\boldsymbol{a}_{(l)}\}$ by standard forward propagation

3: **for** layer $l = \cdots, 2, 1$ **do**
4:     Get output gradient $\{\frac{\partial L}{\partial \boldsymbol{s}_{(l)}}\}$ by standard backward propagation
5:     Compute the layer-wise per-example gradient norm $\|\frac{\partial L_i}{\partial \mathbf{W}_{(l)}}\|^2$
6:     **if** $l$ is the first layer of the $m$-th group $\mathcal{G}_m$ **then**
7:         Aggregate gradient norm across layers in this group: $\|\frac{\partial L_i}{\partial \mathbf{W}^{(m)}}\|^2 = \sum_{r \in \mathcal{G}_m} \|\frac{\partial L_i}{\partial \mathbf{W}_{(r)}}\|^2$
8:         Compute group-wise per-sample clipping factor: $C_i^{(m)} = 1/(\|\frac{\partial L_i}{\partial \mathbf{W}^{(m)}}\| + 0.01)$
9:         **for** layer $r \in \mathcal{G}_m$ **do**
10:             Compute sum of clipped gradients $\mathbf{G}_r = \sum_i \frac{\partial C_i^{(m)} L_i}{\partial \mathbf{W}_{(r)}} = \boldsymbol{a}_{(r)}^\top \mathrm{diag}(C_i^{(m)}) \frac{\partial L}{\partial \boldsymbol{s}_{(r)}}$
11:             Delete $\{\boldsymbol{a}_{(r)}\}, \{\frac{\partial L}{\partial \boldsymbol{s}_{(r)}}\}$

12: Apply SGD/Adam/LAMB with the private gradient $\mathbf{G}_{\mathrm{private}} = \mathbf{G} + \sigma_{\mathrm{DP}} \cdot \mathcal{N}(0, \mathbf{I})$

---

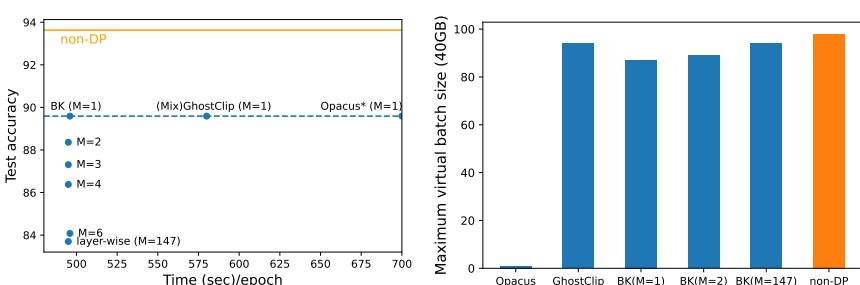

Figure 3: Accuracy and memory of ViT-large on CIFAR100 ($\epsilon = 2$), with uniform group-wise clipping styles. Virtual batch size is 50 except for Opacus, which incurs OOM error (requiring more than 150GB) and at most uses virtual batch size 1 (more than 8 hours per epoch).

In particular, we emphasize that *all group-wise clippings have the same time complexity* by the BK algorithm, since they only differ in the ordering of computation of output gradients and parameter gradients. This is visualized by the same height in Figure 2 and verified in Figure 3 and Figure 5.

## 4 CONVERGENCE OF DP-SGD WITH GROUP-WISE CLIPPING

In this section, we prove the high-probability convergence with any group-wise clipping styles, e.g. layer-wise, parameter-wise, block-wise, all-layer and so on. Our proof only relies on common assumptions in the literature of standard non-DP SGD[4].

**Assumption 4.1** (Lower bound of loss). For all $\mathbf{w}$, we have $L(\mathbf{w}) \geq L_*$ for some constant $L_*$.

**Assumption 4.2** (Smoothness). Let $\boldsymbol{g}(\mathbf{w})$ denote the gradient of the objective $\mathcal{L}(\mathbf{w})$. Then $\forall \mathbf{w}, \boldsymbol{v}$, there is an non-negative constant $\mathcal{L}$ such that

$$L(\boldsymbol{v}) - [L(\mathbf{w}) + \boldsymbol{g}(\mathbf{w})^\top (\boldsymbol{v} - \mathbf{w})] \leq \frac{\mathcal{L}}{2}\|\mathbf{w} - \boldsymbol{v}\|^2 \tag{5}$$

---

[3]We adopt the BK algorithm which is much faster than its alternatives – GhostClip Li et al. (2021); Bu et al. (2022a) and OpacusYousefpour et al. (2021).

[4]The symmetric gradient noise in Assumption 4.3 is widely used for mini-batched SGD analysis Mandt et al. (2017); Smith et al. (2018); Chaudhari & Soatto (2018); Xie et al. (2020), which reduces to per-sample gradient when batch size is 1.

**Assumption 4.3** (Gradient noise). The per-sample gradient noise $\boldsymbol{g}_{t,i}^{(m)} - \boldsymbol{g}_t^{(m)}$ is i.i.d. such that

$$\mathbb{E}(\boldsymbol{g}_{t,i}^{(m)} - \boldsymbol{g}_t^{(m)}) = 0, \mathbb{E}\|\boldsymbol{g}_{t,i}^{(m)} - \boldsymbol{g}_t^{(m)}\|^2 \leq \xi^2/M,$$

and $\boldsymbol{g}_{t,i}^{(m)}$ is symmetric about the oracle gradient $\boldsymbol{g}_t^{(m)}$: $\boldsymbol{g}_{t,i}^{(m)} - \boldsymbol{g}_t^{(m)} \stackrel{\mathcal{D}}{=} \boldsymbol{g}_t^{(m)} - \boldsymbol{g}_{t,i}^{(m)}$.

### 4.1 EFFECT OF NUMBER OF GROUPS ON THE CONVERGENCE

**Theorem 1.** *Under Assumption 4.1, 4.2, 4.3, running DP-SGD with AUTO group-wise clipping* (4) *for $T$ iterations gives, for arbitrarily small and positive $\varrho$:*

$$\max_t \mathbb{P}\left(\|\boldsymbol{g}_t\| < O(\varrho^{-3/2}T^{-1/4})\right) \geq 1 - \varrho \tag{6}$$

*where $\boldsymbol{g}_t = [\boldsymbol{g}_t^{(1)}, \cdots, \boldsymbol{g}_t^{(M)}]$ and $O(\varrho^{-3/2}T^{-1/4}) = (20\xi + \frac{\sqrt{M}}{5})\left(\frac{2M(L_0 - L_*)\mathcal{L}\left(1 + \frac{\sigma^2 d}{B^2}\right)}{\varrho T}\right)^{\frac{1}{4}} + O\left(\varrho^{-\frac{3}{2}}T^{-\frac{3}{4}}\right)$. Note that the result for the all-layer clipping corresponds to $M = 1$.*

*In contrast, running the standard (non-DP) SGD for $T$ iterations gives:*

$$\max_t \mathbb{P}\left(\|\boldsymbol{g}_t\| < O(\varrho^{-1}T^{-1/4})\right) \geq 1 - \varrho. \tag{7}$$

*where $O(\varrho^{-1}T^{-1/4}) = \frac{1}{\varrho T^{1/4}}\sqrt{2(L_0 - L_*)\mathcal{L} + \frac{\xi^2}{B}}$.*

We observe in (6) that partitioning trainable parameters into more groups (larger $M$) negatively affects the convergence guarantee. This is empirically verified in Figure 1 and Section 7 across various models. We note, this result does not contradict the fact that the layer-wise clipping uses a finer grouping than the all-layer clipping.

### 4.2 FINER GROUPING DOES NOT NECESSARILY IMPLY MORE ACCURATE CLIPPING

Suppose we have two groupings: $\mathcal{G}_2$ is finer than $\mathcal{G}_1$, in the sense that each group in $\mathcal{G}_1$ is partitioned into more groups in $\mathcal{G}_2$ (notice that a finer grouping has more groups but the converse does not always hold). It may be tempting to expect that, with the optimal tuning of clipping threshold, the finer grouping is at least as good as the other.

Somewhat surprisingly, we show that this is not the case: it is not true that the group-wise clipping based on $\mathcal{G}_1$ is a subset of that based on $\mathcal{G}_2$. For instance, the all-layer clipping cannot be viewed as a sub-case of the layer-wise clipping. We prove in Theorem 2 with counter-examples that hold for both Abadi's and AUTO clipping functions.

**Theorem 2.** *Consider 2 layers of parameters. There exist per-sample gradients $\boldsymbol{g}_i, \boldsymbol{g}_j$, such that the all-layer clipping $\mathcal{G}_1 = \{1, 2\}$ cannot be represented as any group-wise clipping $\mathcal{G}_2 = \{1\}, \{2\}$: $\exists R, \forall(R_1, R_2)$, at least one of the following holds,*

$$\boldsymbol{g}_i \cdot C(\boldsymbol{g}_i; R_1, R_2) \neq \boldsymbol{g}_i \cdot C(\boldsymbol{g}_i; R)$$

$$\boldsymbol{g}_j \cdot C(\boldsymbol{g}_j; R_1, R_2) \neq \boldsymbol{g}_j \cdot C(\boldsymbol{g}_j; R)$$

*Proof.* We demonstrate with Abadi's clipping function, though AUTO clipping function can be similarly analyzed. Consider $\boldsymbol{g}_i = [3, 4], \boldsymbol{g}_j = [6, 0]$, and $R = 4$. By all-layer clipping, $\langle 1 \rangle\, \boldsymbol{g}_i$ is clipped, so are its components $\boldsymbol{g}_i^{(1)}, \boldsymbol{g}_i^{(2)}$; $\langle 2 \rangle\, \boldsymbol{g}_j$ is not clipped, neither are its components $\boldsymbol{g}_j^{(1)}, \boldsymbol{g}_j^{(2)}$. No matter how one chooses $R_1$, it's impossible to reproduce the same clipped per-sample gradients: if $R_1 < \|\boldsymbol{g}_j^{(1)}\|$, contradicting $\langle 2 \rangle$; otherwise, $R_1 > \|\boldsymbol{g}_j^{(1)}\| > \|\boldsymbol{g}_i^{(1)}\|$, contradicting $\langle 1 \rangle$. $\square$

In fact, we show through the extended proof in appendix that, it is almost impossible in practice that all-layer clipping is a subset of layer-wise clipping. Therefore, one cannot guarantee that the optimal layer-wise clipping (even if adaptively tuned at every iteration) has higher accuracy than the optimal all-layer clipping.

## 5 UNIFORM V.S. NON-UNIFORM DESIGN OF GROUPING

Besides the number of groups, the design of grouping is also critical to the performance of group-wise clipping, although each group-wise clipping is equally $(\epsilon, \delta)$-DP if $\sigma_{\mathrm{DP}}$ is fixed in (1).

**Remark 5.1.** The grouping of trainable parameters does not affect the privacy guarantee, because the noise-to-sensitivity ratio is the same (McMahan et al., 2018b).

Next, we claim that exhausting all the grouping of layers is computationally infeasible, because the possibilities of grouping is known as the Bell number (Bell, 1938). This number grows faster than exponentially, with $10^{4.3}$ grouping of 9 layers and $10^{11.8}$ grouping of 18 layers.

Hence, we seek interesting sub-space of the grouping by investigating many factors that justify a good grouping. Existing designs of group-wise clipping, like layer-wise and per-device clipping, are uniform in the sense that each group has roughly the same number of layers or parameters. In Figure 1, we uniformly group the transformer blocks by the common divisor $\{2,3,4,6\}$ because ViT and RoBERTa have either 12 or 24 blocks.

While the uniform grouping is easy to design, we explore the non-uniform grouping as a broader class that contains uniform ones as special cases, which partitions different number of layers in each group. As we will discuss in Section 6, the non-uniform group-wise clipping can reach beyond the accuracy-memory trade-off of the uniform one.

## 6 PEAK MEMORY PROFILING

Different grouping has different memory profile, especially in terms of the maximum peak memory in Table 3. As its name suggests, BK algorithm (originally proposed only with all-layer clipping style) book-keeps the output gradients across all layers, which results in a high peak memory[5].

Table 3: Accuracy and maximum peak memory of two-group clipping style. Here 'boundary' means the first X attention blocks are the first group and the other (12-X) blocks are the second group.

| boundary | CIFAR100 ViT-large | | QNLI RoBERTa-base | |
|---|---|---|---|---|
| | test accuracy | peak memory (GB) | test accuracy | peak memory (GB) |
| 2 | 88.06 | 22.04 | 85.67 | 11.94 |
| 4 | 88.27 | 21.96 | 85.92 | 11.95 |
| 6(uniform) | 88.36 | 21.89 | 86.22 | 12.22 |
| 8 | 88.75 | 21.82 | 86.38 | 13.40 |
| 10 | 88.89 | 21.75 | 86.29 | 14.57 |
| all-layer | 89.59 | 23.52 | 86.91 | 15.75 |
| non-DP | 93.63 | 20.38 | 87.34 | 11.63 |

For the group-wise clipping style, we can characterize the optimization's memory profile by the memory peaks: when back-propagation arrives at the first layer of the $m$-th group, at Line 6 in Algorithm 1, all output gradients in this group and all activation tensors in the un-processed groups are cached in the memory. Therefore, $M$ groups lead to $M$ memory peaks, whose form in Fact 6.1 is proved in appendix.

**Fact 6.1.** The $m$-th memory peak by space complexity is

$$B\Big( \sum_{l<\mathcal{G}_m[-1]} T_l d_l + \sum_{\mathcal{G}_m[0]<r<\mathcal{G}_m[-1]} T_r p_r \Big).$$

Hence, we define the maximum memory peak as

$$B \max_m \Big\{ \sum_{l<\mathcal{G}_m[-1]} T_l d_l + \sum_{\mathcal{G}_m[0]<r<\mathcal{G}_m[-1]} T_r p_r \Big\} \tag{8}$$

---

[5]For GPT2 models, DP optimization with all-layer clipping incurs 30% more memory cost than non-DP by (Bu et al., 2022c, Table 8), when the sentence length is long and hence the output gradients are expensive.

which negatively determines the maximum (virtual) batch size[6], and thus the maximum throughput (i.e. training speed) of the DP optimization.

## 6.1 MAXIMUM PEAK MEMORY OF UNIFORM GROUPING

For uniform grouping, the maximum memory peak is always that of the bottom group (the firstly processed during back-propagation), i.e. $m = M$. We visualize the memory peaks of in Figure 4, where the group-wise clipping is $M = 4$, and the layer-wise clipping is $M = 147$.

We highlight that the maximum peak memory of all-layer clipping occurs when the back-propagation reaches the top layer since all output gradients are book-kept. For layer-wise clipping, the maximum peak memory is similar to that of non-DP training (see also Figure 1), whose peak memory occurs when the back-propagation just starts.

Generally speaking, for uniform grouping, the peak memory increases with smaller number of groups $M$, though the throughput is not affected under BK algorithm (see Figure 5, explained by Section 3).

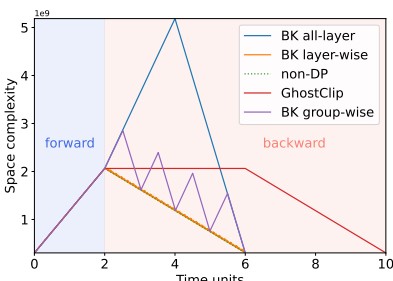

Figure 4: Space complexity of forward and backward propagation. We use ViT-large-patch16-224 with a batch size of 32.

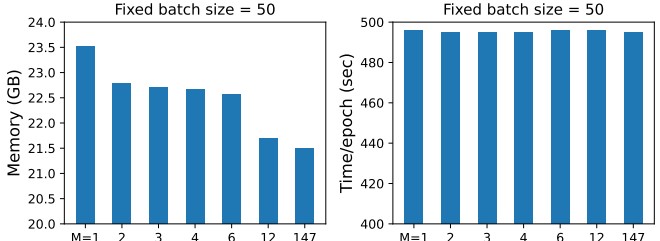

Figure 5: Peak memory and throughput of ViT-large-patch16-224 using fixed batch size.

## 6.2 MAXIMUM PEAK MEMORY OF NON-UNIFORM GROUPING

The maximum peak memory of non-uniform grouping can still be described by (8), but not as explicitly as the uniform grouping. For example, with the two-group clipping style, the maximum peak memory may be the first peak or the second one. This explains the non-monotone pattern in Table 3, and motivates to group layers so that the two memory peaks are similar.

Given that the non-uniform grouping contains the uniform grouping as special cases, it usually breaks the accuracy-memory trade-off of the uniform grouping, see Figure 6.

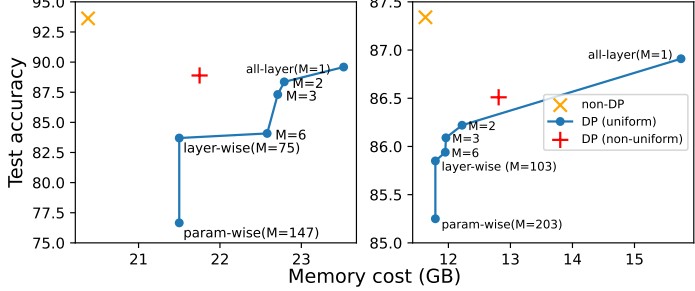

Figure 6: Accuracy and memory on CIFAR100 (ViT-large) and QNLI (RoBERTa-base).

---

[6]Virtual batch size is the number of samples sent to computing devices, which is necessary for gradient accumulation and distributed learning. It only affects the training efficiency but not the accuracy, as the latter is determined by the logical batch size.

## 7 EXPERIMENTS

We experiment the group-wise clipping style on multiple vision and language tasks, in Table 4, Table 5 and Table 7 (in appendix). We focus on the uniform grouping, and cover ViT Dosovitskiy et al. (2020) / RoBERTa Liu et al. (2019) / GPT2 Radford et al. models. Empirically speaking, more groups indicate worse accuracy than the all-layer clipping, where the gap decreases as the privacy budget and the model capacity increases. For example, the accuracy gap between layer-wise and all-layer clipping drops from 10% ($\epsilon = 2$) to 5% ($\epsilon = 8$), when training ViT-small on CIFAR100, and further drops to 2% when training ViT-large. We observe the similar patterns on the text datasets with RoBERTa and GPT2, in which the performance of layer-wise clipping is comparable to that of all-layer clipping. Specially, our results are comparable with the adaptive layer-wise clipping (denoted as ∗ in Table 7 and Table 5) He et al. (2022), even though they trained 20 epochs on SST2 but we only train 3 epochs.

Table 4: Test accuracy of image classification tasks under group-wise clipping styles.

| Model | Method | $\epsilon = 2$ | | | | | $\epsilon = 8$ | | | | |
|---|---|---|---|---|---|---|---|---|---|---|---|
| | | CIFAR10 | CIFAR100 | SVHN | GTSRB | Food101 | CIFAR10 | CIFAR100 | SVHN | GTSRB | Food101 |
| ViT -small | all-layer ($M = 1$) | 96.94 | 82.58 | 91.28 | 89.57 | 73.45 | 97.13 | 85.00 | 92.92 | 94.55 | 77.12 |
| | non-uniform ($M = 2$) | 97.01 | 82.68 | 90.77 | 89.70 | 74.45 | 97.11 | 85.23 | 92.24 | 94.77 | 78.14 |
| | uniform ($M = 2$) | 96.87 | 81.79 | 91.39 | 89.55 | 72.48 | 97.00 | 84.45 | 92.98 | 94.59 | 76.68 |
| | uniform ($M = 3$) | 96.81 | 80.96 | 91.02 | 89.55 | 71.97 | 96.99 | 84.36 | 92.71 | 94.70 | 76.37 |
| | uniform ($M = 4$) | 96.81 | 80.45 | 90.95 | 89.70 | 71.47 | 96.96 | 84.09 | 92.52 | 94.66 | 76.14 |
| | uniform ($M = 6$) | 96.72 | 79.24 | 90.96 | 89.89 | 70.88 | 96.93 | 83.49 | 92.44 | 94.67 | 75.76 |
| | layer-wise ($M = 75$) | 96.60 | 71.93 | 90.42 | 87.18 | 65.21 | 96.86 | 80.49 | 91.87 | 93.88 | 71.96 |
| | param-wise ($M = 150$) | 96.34 | 63.24 | 89.04 | 80.77 | 59.42 | 96.71 | 75.09 | 90.90 | 91.91 | 67.85 |
| ViT -large | all-layer ($M = 1$) | 98.68 | 89.59 | 93.27 | 91.81 | 82.29 | 98.92 | 90.66 | 94.26 | 95.68 | 84.84 |
| | non-uniform ($M = 2$) | 98.60 | 88.89 | 93.14 | 91.61 | 81.46 | 98.90 | 90.36 | 94.16 | 95.70 | 84.39 |
| | uniform ($M = 2$) | 98.52 | 88.36 | 92.77 | 90.89 | 81.13 | 98.69 | 90.36 | 93.79 | 95.27 | 84.05 |
| | uniform ($M = 3$) | 98.59 | 87.31 | 92.42 | 90.31 | 79.68 | 98.71 | 89.95 | 93.59 | 95.17 | 83.33 |
| | uniform ($M = 4$) | 98.51 | 86.38 | 92.31 | 89.97 | 78.46 | 98.70 | 89.48 | 93.34 | 94.88 | 82.61 |
| | uniform ($M = 6$) | 98.56 | 84.08 | 92.16 | 88.99 | 76.66 | 98.66 | 88.89 | 93.19 | 94.73 | 81.59 |
| | layer-wise ($M = 147$) | 98.37 | 83.70 | 92.61 | 89.87 | 77.89 | 98.57 | 88.65 | 93.79 | 94.62 | 82.68 |
| | param-wise ($M = 294$) | 98.24 | 76.66 | 91.48 | 85.28 | 72.59 | 98.47 | 86.43 | 93.14 | 92.73 | 79.61 |

Table 5: Test score of text generation on E2E dataset under group-wise clipping styles.

| Model | Method | | $\epsilon = 3$ | | | | | $\epsilon = 8$ | | | | |
|---|---|---|---|---|---|---|---|---|---|---|---|---|
| | | | BLEU | ROGUE-L | NIST | METEOR | CIDEr | BLE | ROGUE-L | NIST | METEOR | CIDEr |
| GPT2 -small | RGP | (Yu et al., 2021b) | 58.48 | 65.56 | 5.775 | 0.331 | 1.300 | 58.46 | 65.03 | 6.276 | 0.349 | 1.496 |
| | all-layer ($M = 1$) | Li et al. (2021) | 61.52 | 65.67 | 6.697 | 0.384 | 1.761 | 63.19 | 66.43 | 7.444 | 0.400 | 1.919 |
| | all-layer ($M = 1$) | Bu et al. (2022b) | 61.34 | 65.87 | 7.071 | 0.387 | 1.801 | 63.60 | 67.07 | 7.714 | 0.404 | 1.938 |
| | block-wise ($M = 12$) | ours | 61.03 | 66.07 | 6.863 | 0.388 | 1.787 | 63.65 | 67.36 | 7.773 | 0.406 | 1.951 |
| | layer-wise ($M = 76$) | ours | 60.76 | 65.93 | 6.680 | 0.386 | 1.766 | 63.47 | 67.49 | 7.791 | 0.407 | 1.975 |
| | layer-wise ($M = 76$) | He et al. (2022) | 61.10 | 65.12 | - | - | - | 63.42 | 66.69 | - | - | - |
| | param-wise ($M = 149$) | ours | 57.97 | 64.84 | 6.002 | 0.372 | 1.624 | 62.07 | 66.27 | 7.197 | 0.393 | 1.848 |
| GPT2 -medium | all-layer ($M = 1$) | Bu et al. (2022b) | 63.85 | 67.07 | 7.106 | 0.387 | 1.754 | 64.22 | 67.53 | 8.172 | 0.418 | 2.081 |
| | block-wise ($M = 24$) | ours | 61.43 | 66.93 | 7.998 | 0.411 | 2.009 | 64.07 | 68.18 | 8.332 | 0.429 | 2.230 |
| | layer-wise ($M = 148$) | ours | 61.80 | 66.76 | 7.865 | 0.407 | 1.974 | 63.96 | 68.44 | 8.325 | 0.429 | 2.237 |
| | param-wise ($M = 293$) | ours | 60.48 | 64.95 | 6.981 | 0.391 | 1.804 | 62.50 | 67.28 | 8.178 | 0.419 | 2.098 |
| GPT2 -large | all-layer ($M = 1$) | Bu et al. (2022b) | 64.18 | 67.86 | 7.937 | 0.403 | 2.008 | 64.64 | 68.97 | 8.301 | 0.420 | 2.163 |
| | block-wise ($M = 36$) | ours | 65.23 | 69.04 | 8.467 | 0.435 | 2.234 | 66.90 | 69.87 | 8.548 | 0.444 | 2.355 |
| | layer-wise ($M = 220$) | ours | 64.86 | 68.30 | 8.417 | 0.431 | 2.219 | 66.44 | 69.55 | 8.504 | 0.443 | 2.297 |
| | param-wise ($M = 437$) | ours | 63.85 | 67.83 | 8.303 | 0.416 | 2.093 | 65.01 | 68.79 | 8.429 | 0.435 | 2.238 |

## 8 DISCUSSION

We show that group-wise clipping, a superset that covers existing clipping styles, leads to different accuracy and efficiency depending on the grouping of trainable parameters. For accuracy, a small number of groups (e.g. the all-layer clipping) benefits the convergence, though the accuracy gap among different group-wise clippings is smaller for larger models. For time efficiency, all group-wise clippings are equally fast under the BK algorithm. For memory efficiency, the uniform group-wise clipping with more groups has smaller peak memory and thus form an accuracy-memory trade-off. However, the non-uniform grouping can reach beyond this trade-off with a careful design. Overall, a proper group-wise clipping style makes system design easy and allows large models to be accurate, fast to train, and memory-efficient. Thus, we establish new state-of-the-art results on multiple datasets, without relying on adaptive clipping or longer training epochs. For future work, more exploration of the grouping is desirable, especially in the orthogonal direction of parameter-efficient fine-tuning.

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

## A  PROOFS

### A.1  PROOF OF THEOREM 2

*Proof.* (**Proof for Abadi's clipping**) Consider a two-layer neural network and two per-sample gradients $\boldsymbol{g}_1 = [\boldsymbol{g}_1^{(1)}, \boldsymbol{g}_1^{(2)}], \boldsymbol{g}_2 = [\boldsymbol{g}_2^{(1)}, \boldsymbol{g}_2^{(2)}]$. Suppose $\|\boldsymbol{g}_1^{(1)}\| < \|\boldsymbol{g}_2^{(1)}\|$ and $\|\boldsymbol{g}_1\| > R > \|\boldsymbol{g}_2\|$. Then in the all-layer clipping, the first per-sample gradient $\boldsymbol{g}_1$ is clipped/scaled, and so are both its components $\boldsymbol{g}_1^{(1)}, \boldsymbol{g}_1^{(2)}$. But the second per-sample gradient $\boldsymbol{g}_2$ is not clipped. However, in the layer-wise clipping with any choice of $\boldsymbol{R} = (R_1, R_2)$, there are two cases both leading to the contradiction.

1. $R_1 < \|\boldsymbol{g}_2^{(1)}\|$. Then $\boldsymbol{g}_2^{(1)}$ is clipped by layer-wise. But $\boldsymbol{g}_2^{(1)}$ is not clipped by all-layer. Done.

2. $R_1 > \|\boldsymbol{g}_2^{(1)}\|$. Then $\boldsymbol{g}_2^{(1)}$ is not clipped by layer-wise, and so is not $\boldsymbol{g}_1^{(1)}$ since $\|\boldsymbol{g}_1^{(1)}\| < \|\boldsymbol{g}_2^{(1)}\|$. But $\boldsymbol{g}_1^{(1)}$ is clipped by all-layer. Done.

In fact, we can generalize this non-equivalence between the all-layer and layer-wise clipping: for small $R$ (say $R < \|\boldsymbol{g}_i\|$), a necessary (but impossible) condition to claim that, the all-layer clipping is a sub-case of the layer-wise clipping, would be

$$\frac{\|\boldsymbol{g}_i^{(m)}\|}{\|\boldsymbol{g}_i\|} = \frac{\|\boldsymbol{g}_j^{(m)}\|}{\|\boldsymbol{g}_j\|}, \forall i, j \in [B], m \in [M].$$

Here $B$ is the batch size and $M$ is the number of groups.

(**Proof for automatic clipping**) Here we work with the $R$-dependent automatic clipping, which is mathematically equivalent to the automatic clipping in (4), according to Theorem 1& 2 in Bu et al. (2022b). We consider a $M$-group neural network and per-sample gradients $\boldsymbol{g}_i$. Suppose the layer-wise clipping can represent the all-layer clipping with $R$, then

$$\frac{R_m \boldsymbol{g}_i^{(m)}}{\|\boldsymbol{g}_i^{(m)}\| + 0.01} = \frac{R \boldsymbol{g}_i^{(m)}}{\|\boldsymbol{g}_i\| + 0.01}, \forall i \in [B], m \in [M].$$

This requires that

$$\frac{\|\boldsymbol{g}_i^{(m)}\| + 0.01}{\|\boldsymbol{g}_i\| + 0.01} = \frac{\|\boldsymbol{g}_j^{(m)}\| + 0.01}{\|\boldsymbol{g}_j\| + 0.01}, \forall i, j \in [B], m \in [M],$$

which is practically impossible to hold. □

### A.2  PROOF OF THEOREM 1: DP-SGD

*Proof.* Consider DP-SGD with the automatic (AUTO-S Bu et al. (2022b)) clipping in a layer-wise style, i.e. $\gamma = 0.01$.

$$\mathbf{w}_{t+1}^{(m)} = \mathbf{w}_t^{(m)} - \eta \left( \frac{\sum_i \boldsymbol{g}_{t,i}^{(m)} / (\|\boldsymbol{g}_{t,i}^{(m)}\| + \gamma)}{\sqrt{M}} + \sigma \mathcal{N}(0, \mathbf{I}) \right)$$

where $\boldsymbol{g}_{t,i}^{(m)}$ is i.i.d. unbiased estimate of $\boldsymbol{g}_t^{(m)}$, with a bounded variance as described in Assumption 4.3.

By the Lipschitz smoothness in Assumption 4.2,

$$L_{t+1} - L_t \leq \sum_m \left[ \boldsymbol{g}_t^{(m)\top} (\mathbf{w}_{t+1}^{(m)} - \mathbf{w}_t^{(m)}) \right] + \frac{\mathcal{L}}{2} \sum_m \|\mathbf{w}_{t+1}^{(m)} - \mathbf{w}_t^{(m)}\|^2$$

$$= -\eta \sum_m \left[ \boldsymbol{g}_t^{(m)\top} \left( \sum_i \frac{\boldsymbol{g}_{t,i}^{(m)}}{\sqrt{M}(\|\boldsymbol{g}_{t,i}^{(m)}\| + \gamma)} + \sigma \mathcal{N}(0, I) \right) \right] + \frac{\mathcal{L}\eta^2}{2} \sum_m \left\| \sum_i \frac{\boldsymbol{g}_{t,i}^{(m)}}{\sqrt{M}(\|\boldsymbol{g}_{t,i}^{(m)}\| + \gamma)} + \sigma \cdot \mathcal{N}(0, \mathbf{I}) \right\|^2$$

Given the fact that $\left\|\frac{\boldsymbol{g}_{t,i}^{(m)}}{\|\boldsymbol{g}_{t,i}^{(m)}\|+\gamma}\right\| \leq 1$, we expand the square of norm and the expected improvement at one iteration is

$$\mathbb{E}(L_{t+1} - L_t|\mathbf{w}_t) \leq -\frac{\eta}{\sqrt{M}} \sum_m \boldsymbol{g}_t^{(m)\top} \mathbb{E}\left(\sum_i \frac{\boldsymbol{g}_{t,i}^{(m)}}{\|\boldsymbol{g}_{t,i}^{(m)}\|+\gamma}\right) + \frac{\mathcal{L}\eta^2}{2} \sum_m \left(\frac{1}{M}\mathbb{E}\left\|\sum_i \frac{\boldsymbol{g}_{t,i}^{(m)}}{\|\boldsymbol{g}_{t,i}^{(m)}\|+\gamma}\right\|^2 + \sigma^2 d^{(m)}\right)$$

$$\leq -\frac{B\eta}{\sqrt{M}} \sum_m \boldsymbol{g}_t^{(m)\top} \mathbb{E}\left(\frac{\boldsymbol{g}_{t,i}^{(m)}}{\|\boldsymbol{g}_{t,i}^{(m)}\|+\gamma}\right) + \frac{\mathcal{L}\eta^2}{2} \sum_m \left(\frac{B^2}{M} + \sigma^2 d^{(m)}\right)$$

$$(9)$$

in which $d^{(m)}$ is the number of parameters in the $m$-th group and $d = \sum_m d^{(m)}$ is the total number of model parameters.

Now we can lower bound $\boldsymbol{g}_t^{(m)\top}\mathbb{E}\left(\frac{\boldsymbol{g}_{t,i}^{(m)}}{\|\boldsymbol{g}_{t,i}^{(m)}\|+\gamma}\right)$ in (9) by Lemma A.1.

**Lemma A.1.** *Denoting $\|\boldsymbol{g}_t^{(m)}\| - \frac{\xi}{r\sqrt{M}}$ as $x_r$, then for any $r > 1$ we have*

$$\boldsymbol{g}_t^{(m)\top}\mathbb{E}\left(\frac{\boldsymbol{g}_{t,i}^{(m)}}{\|\boldsymbol{g}_{t,i}^{(m)}\|+\gamma}\right) \geq \frac{1}{2} \cdot \underbrace{x_r\left(\frac{\gamma}{(r-1)(x_r+\frac{\xi}{r\sqrt{M}})+\gamma} - \frac{\gamma}{(r+1)(x_r+\frac{\xi}{r\sqrt{M}})+\gamma}\right)}_{\mathcal{M}(x_r;r,\xi,\gamma)}$$

$$(10)$$

*Here $\mathcal{M}$ is non-negative and strictly increasing, with $\mathcal{M}(0) = 0$. Thus $\mathcal{M}$ can be viewed as a distance measure.*

Using this lower bound, the expected improvement (9) becomes

$$\mathbb{E}(L_{t+1} - L_t|\mathbf{w}_t) \leq -\frac{B\eta}{2\sqrt{M}} \sum_m \left[\mathcal{M}(\|\boldsymbol{g}_t^{(m)}\| - \frac{\xi}{r\sqrt{M}})\right] + \frac{\mathcal{L}\eta^2}{2}\left(B^2 + \sigma^2 d\right)$$

Now extend the expectation over randomness in the trajectory, and perform a telescoping sum over the iterations

$$L_0 - L_* \geq L_0 - \mathbb{E}L_T = \sum_t \mathbb{E}(L_t - L_{t+1})$$

$$\geq \frac{B\eta}{2\sqrt{M}}\mathbb{E}\left(\sum_{t,m} \mathcal{M}(\|\boldsymbol{g}_t^{(m)}\| - \frac{\xi}{r\sqrt{M}})\right) - \frac{T\mathcal{L}\eta^2}{2}\left(B^2 + \sigma^2 d\right)$$

Substituting $\eta B = \eta_0/\sqrt{T}$ where $\eta_0$ is a base learning rate, we have

$$2(L_0 - L_*) \geq \sqrt{\frac{T}{M}}\eta_0\mathbb{E}\left(\frac{1}{T}\sum_{t,m} \mathcal{M}(\|\boldsymbol{g}_t^{(m)}\| - \frac{\xi}{r\sqrt{M}})\right) - \mathcal{L}\eta_0^2\left(1 + \frac{\sigma^2 d}{B^2}\right)$$

and finally

$$\mathbb{E}\left(\frac{1}{T}\sum_{t,m} \mathcal{M}(\|\boldsymbol{g}_t^{(m)}\| - \frac{\xi}{r\sqrt{M}})\right) \leq \sqrt{\frac{M}{T}}\left[\frac{2(L_0 - L_*)}{\eta_0} + \mathcal{L}\eta_0\left(1 + \frac{\sigma^2 d}{B^2}\right)\right] \quad (11)$$

With $\eta_0$ chosen properly as $\sqrt{\frac{2(L_0-L_*)}{\mathcal{L}(1+\frac{\sigma^2 d}{B^2})}}$, the hyperbola on the right hand side in (11) is minimized to $2\sqrt{\frac{M}{T}}\sqrt{2(L_0 - L_*)\mathcal{L}\left(1 + \frac{\sigma^2 d}{B^2}\right)}$.

Since the minimum of a sequence is smaller than the average, we have

$$\min_t \mathbb{E}(\sum_m \mathcal{M}(\|\boldsymbol{g}_t^{(m)}\| - \frac{\xi}{r\sqrt{M}})) \leq 2\sqrt{\frac{M}{T}} \sqrt{2(L_0 - L_*)\mathcal{L}\left(1 + \frac{\sigma^2 d}{B^2}\right)} \tag{12}$$

Then by the Markov's inequality (since $\mathcal{M}$ is non-negative), for any constant $a > 0$,

$$\min_t \sum_m \mathbb{P}(\mathcal{M}(\|\boldsymbol{g}_t^{(m)}\| - \frac{\xi}{r\sqrt{M}}) > a) \leq \frac{2}{a}\sqrt{\frac{M}{T}} \sqrt{2(L_0 - L_*)\mathcal{L}\left(1 + \frac{\sigma^2 d}{B^2}\right)} \tag{13}$$

Note that

$$\sum_m \mathbb{P}\left(\mathcal{M}(\|\boldsymbol{g}_t^{(m)}\| - \frac{\xi}{r\sqrt{M}}) > a\right) > 1 - \mathbb{P}\left(\bigcap_m \mathcal{M}(\|\boldsymbol{g}_t^{(m)}\| - \frac{\xi}{r\sqrt{M}}) < a\right)$$

which leads Equation (13) to

$$\max_t \mathbb{P}\left(\bigcap_m \mathcal{M}(\|\boldsymbol{g}_t^{(m)}\| - \frac{\xi}{r\sqrt{M}}) < a\right) \geq 1 - \frac{2\sqrt{M}}{a\sqrt{T}}\sqrt{2(L_0 - L_*)\mathcal{L}\left(1 + \frac{\sigma^2 d}{B^2}\right)} \tag{14}$$

Denoting the inverse function of $\mathcal{M}$ as $\mathcal{M}^{-1}$, whose explicit formula will be given in Lemma A.2, we get

$$\max_t \mathbb{P}\left(\bigcap_m \|\boldsymbol{g}_t^{(m)}\|^2 < \left(\mathcal{M}^{-1}(a) + \frac{\xi}{r\sqrt{M}}\right)^2\right) \geq 1 - \frac{2\sqrt{M}}{a\sqrt{T}}\sqrt{2(L_0 - L_*)\mathcal{L}\left(1 + \frac{\sigma^2 d}{B^2}\right)} \tag{15}$$

It is obvious that $\|\boldsymbol{g}_t^{(m)}\|$ being small for all $1 \leq m \leq M$ is a sufficient condition to guarantee $\|\boldsymbol{g}_t\|$ to be small. Therefore,

$$\mathbb{P}\left(\|\boldsymbol{g}_t\| < \sqrt{M}\left(\mathcal{M}^{-1}(a) + \frac{\xi}{r\sqrt{M}}\right)\right) \geq \mathbb{P}\left(\bigcap_m \|\boldsymbol{g}_t^{(m)}\|^2 < \left(\mathcal{M}^{-1}(a) + \frac{\xi}{r\sqrt{M}}\right)^2\right)$$

and consequently we have the high probability bound for any $r > 1, a > 0$:

$$\max_t \mathbb{P}\left(\|\boldsymbol{g}_t\| < \sqrt{M}\mathcal{M}^{-1}(a; r, \xi, \gamma) + \frac{\xi}{r}\right) \geq 1 - \frac{2\sqrt{M}}{a\sqrt{T}}\sqrt{2(L_0 - L_*)\mathcal{L}\left(1 + \frac{\sigma^2 d}{B^2}\right)}. \tag{16}$$

In order for $\|\boldsymbol{g}_t\|$ to converge to zero, we need both $\mathcal{M}^{-1}(a) \to 0$ and $\frac{\xi}{r} \to 0$, as $T \to \infty$. I.e. we consider $a \to 0$. We use Lemma A.2 to claim that, under any fixed $r$,

$$\sqrt{M}\mathcal{M}^{-1}(a) + \frac{\xi}{r} = r \cdot \frac{aM(\frac{\xi}{\sqrt{M}} + \gamma)^2}{2\xi\gamma} + \frac{1}{r} \cdot \left(\xi - \frac{a\xi}{2\gamma}\right) + o(a)$$

so that

$$\min_r \sqrt{M}\mathcal{M}^{-1}(a) + \frac{\xi}{r} = 2\sqrt{\frac{aM(\frac{\xi}{\sqrt{M}} + \gamma)^2}{2\xi\gamma} \cdot \left(\xi - \frac{a\xi}{2\gamma}\right)} + o(a)$$

where the last equality is obvious for a hyperbola with respect to $r$. In fact, the square root term simplifies to $\sqrt{2a(\xi + \gamma\sqrt{M})^2/\gamma + O(a^{1.5})}$, and so does the whole term. To put this into perspective, we denote $\varrho := \frac{2\sqrt{M}}{a\sqrt{T}}\sqrt{2(L_0 - L_*)\mathcal{L}\left(1 + \frac{\sigma^2 d}{B^2}\right)}$. Then we can write Equation (16) asymptotically

$$\max_t \mathbb{P}\left(\|\boldsymbol{g}_t\| < \sqrt{2a(\xi + \gamma\sqrt{M})^2/\gamma + O(a^{1.5})}\right) \geq 1 - \varrho.$$

which becomes

$$\max_t \mathbb{P}\left(\|\boldsymbol{g}_t\| < 2(\xi + \gamma\sqrt{M})\sqrt{\frac{\sqrt{M}}{\varrho\gamma}\sqrt{\frac{2(L_0 - L_*)\mathcal{L}\left(1 + \frac{\sigma^2 d}{B^2}\right)}{T}}} + O\left(\frac{1}{\varrho^{1.5}T^{0.75}}\right)\right) \geq 1 - \varrho.$$

$$\square$$

**Lemma A.2.** *The explicit form of $\mathcal{M}^{-1}$ is*

$$\mathcal{M}^{-1}(x; r, \xi, \gamma) = \frac{-\frac{\xi}{r\sqrt{M}}\gamma + (r^2 - 1)\frac{\xi}{r\sqrt{M}}x + r\gamma x + \gamma\sqrt{(\frac{\xi}{r\sqrt{M}})^2 + 2\frac{\xi}{\sqrt{M}}x + 2\gamma x + x^2}}{2\gamma - (r^2 - 1)x},$$
(17)

*and the asymptotic form (as $x \to 0$) is linear:*

$$\mathcal{M}^{-1}(x; r, \xi, \gamma) = x \cdot \frac{r^2(\frac{\xi}{\sqrt{M}} + \gamma)^2 - (\frac{\xi}{\sqrt{M}})^2}{2\frac{\xi}{\sqrt{M}}\gamma r} + o(x),$$
(18)

*Proof of Lemma A.2.* The explicit form of $\mathcal{M}^{-1}$ can be easily verified by $\mathcal{M}^{-1}(\mathcal{M}(x)) = x$. In fact, this has already been shown in Bu et al. (2022b) where $M = 1$ (i.e. we switch $\xi$ to $\xi/\sqrt{M}$). One can also check this using Equation (10) and WolframAlpha by searching

```
inverse function cx/((r-1)(x+a)+c)-cx/((r+1)(x+a)+c)
```

where c means $\gamma$, a means $\frac{\xi}{r\sqrt{M}}$, and r means $r$.

The asymptotic form of $\mathcal{M}^{-1}$ can be derived from the asymptotic form of $\mathcal{M}$ in Equation (10):

$$\mathcal{M}(x) = x\left(\frac{\gamma}{(r-1)\frac{\xi}{r\sqrt{M}} + \gamma} - \frac{\gamma}{(r+1)\frac{\xi}{r\sqrt{M}} + \gamma}\right) + O(x^2) = \frac{2\frac{\xi}{\sqrt{M}}\gamma r x}{r^2(\frac{\xi}{\sqrt{M}} + \gamma)^2 - (\frac{\xi}{\sqrt{M}})^2} + O(x^2).$$

This can be checked by WolframAlpha through

```
x(c/((r-1)(x+a)+c)-c/((r+1)(x+a)+c)) expand x=0
```

Therefore, we have

$$\mathcal{M}^{-1}(x) = x \cdot \frac{r^2(\frac{\xi}{\sqrt{M}} + \gamma)^2 - (\frac{\xi}{\sqrt{M}})^2}{2\frac{\xi}{\sqrt{M}}\gamma r} + O(x^2).$$

□

## A.3 Proof of Theorem 1: standard SGD

*Proof.* This proof is similar to Theorem 4 in Bu et al. (2022b), though theirs is of expected convergence and ours is of high probability. Consider standard (non-DP) SGD,

$$\mathbf{w}_{t+1}^{(m)} = \mathbf{w}_t^{(m)} - \eta\frac{\sum_i \boldsymbol{g}_{t,i}^{(m)}}{B}$$

where $\boldsymbol{g}_{t,i}^{(m)}$ is i.i.d. unbiased estimate of $\boldsymbol{g}_t^{(m)}$ with a bounded variance in Assumption 4.3.

By Lipschitz smoothness in Assumption 4.2,

$$L_{t+1} - L_t \leq \sum_m \left[\boldsymbol{g}_t^{(m)\top}(\mathbf{w}_{t+1}^{(m)} - \mathbf{w}_t^{(m)})\right] + \frac{\mathcal{L}}{2}\sum_m \|\mathbf{w}_{t+1}^{(m)} - \mathbf{w}_t^{(m)}\|^2$$

$$= -\eta\sum_m \left[\boldsymbol{g}_t^{(m)\top}\left(\sum_i \frac{\boldsymbol{g}_{t,i}^{(m)}}{B}\right)\right] + \frac{\mathcal{L}\eta^2}{2}\sum_m \left\|\sum_i \frac{\boldsymbol{g}_{t,i}^{(m)}}{B}\right\|^2.$$

The expected improvement at one iteration is

$$\mathbb{E}(L_{t+1} - L_t|\mathbf{w}_t) \leq -\eta \sum_m \boldsymbol{g}_t^{(m)\top} \mathbb{E}\left(\boldsymbol{g}_{t,i}^{(m)}\right) + \frac{\mathcal{L}\eta^2}{2} \sum_m \mathbb{E}\left(\|\sum_i \frac{\boldsymbol{g}_{t,i}^{(m)}}{B}\|^2\right)$$

$$\leq -\eta \sum_m \boldsymbol{g}_t^{(m)\top} \boldsymbol{g}_t^{(m)} + \frac{\mathcal{L}\eta^2}{2} \sum_m (\|\boldsymbol{g}_t^{(m)}\|^2 + \frac{\xi^2}{LB}) \qquad (19)$$

$$= -\eta\|\boldsymbol{g}_t\|^2 + \frac{\mathcal{L}\eta^2}{2}(\|\boldsymbol{g}_t\|^2 + \frac{\xi^2}{B})$$

Notice that Equation (19) does not require the symmetry assumption in Assumption 4.3 for the per-sample gradient noise. We extend the expectation over randomness in the trajectory, and perform a telescoping sum,

$$L_0 - L_* \geq \sum_t \mathbb{E}(L_t - L_{t+1}) \geq \left(\eta - \frac{\mathcal{L}\eta^2}{2}\right) \mathbb{E}(\sum_t \|\boldsymbol{g}_t\|^2) - \frac{T\mathcal{L}\eta^2\xi^2}{2B}$$

We apply the same learning rate as in Bernstein et al. (2018) and Bu et al. (2022b), namely $\eta = 1/\mathcal{L}\sqrt{T}$, to get

$$L_0 - L_* \geq \left(\frac{1}{\mathcal{L}\sqrt{T}} - \frac{1}{2\mathcal{L}T}\right) \mathbb{E}\left(\sum_t \|\boldsymbol{g}_t\|^2\right) - \frac{\xi^2}{2B\mathcal{L}} > \frac{\sqrt{T}}{2\mathcal{L}} \mathbb{E}\left(\frac{1}{T}\sum_t \|\boldsymbol{g}_t\|^2\right) - \frac{\xi^2}{2B\mathcal{L}}$$

and thus

$$\min_t \mathbb{E}\left(\|\boldsymbol{g}_t\|^2\right) \leq \frac{1}{T} \sum_t \mathbb{E}\left(\|\boldsymbol{g}_t\|^2\right) = \mathbb{E}\left(\frac{1}{T}\sum_t \|\boldsymbol{g}_t\|^2\right) \leq \frac{1}{\sqrt{T}}\left[2(L_0 - L_*)\mathcal{L} + \frac{\xi^2}{B}\right]$$

Using the Jensen's inequality and the Markov's inequality, we can have

$$\min_t a \cdot \mathbb{P}(\|\boldsymbol{g}_t\| > a) \leq \min_t \mathbb{E}\left(\|\boldsymbol{g}_t\|\right) \leq \min_t \sqrt{\mathbb{E}\left(\|\boldsymbol{g}_t\|^2\right)} \leq \frac{1}{T^{1/4}}\sqrt{2(L_0 - L_*)\mathcal{L} + \frac{\xi^2}{B}}$$

for any positive constant $a$. Denoting $a = \frac{1}{T^{1/4}}\sqrt{2(L_0 - L_*)\mathcal{L} + \frac{\xi^2}{B}}/\varrho$, we have

$$\max_t \mathbb{P}\left(\|\boldsymbol{g}_t\| < \frac{1}{\varrho T^{1/4}}\sqrt{2(L_0 - L_*)\mathcal{L} + \frac{\xi^2}{B}}\right) \geq 1 - \varrho.$$

$\square$

## B  EXPERIMENT SETTINGS

All experiments are fully fine-tuned using a single Nvidia A100 GPU.

| Dataset | CIFAR/SVHN/Food101 | GTSRB | MNLI(m/mm) | QQP | QNLI | SST2 | E2E |
|---------|:---:|:---:|:---:|:---:|:---:|:---:|:---:|
| Model | ViT | | RoBERTa | | | | GPT2 |
| Epoch | 5 | 10 | 18 | 18 | 6 | 3 | 10 |
| Batch size | 1000 | 1000 | 12000 | 12000 | 4000 | 2000 | 1000 |
| DP learning rate | 5e-4 | 5e-4 | 3e-4 | 3e-4 | 3e-4 | 3e-4 | 1e-3 |
| learning rate schedule | — | — | — | — | — | — | — |
| AdamW weight decay | 0.01 | 0.01 | 0 | 0 | 0 | 0 | 0.01 |
| Hidden feature dimension | 224*224 | 224*224 | 256 | 256 | 256 | 256 | 100 |

Table 6: Hyperparameters for Table 4, Table 7, and Table 5. Note that we use automatic clipping which need not to set the clipping threshold.

## C  EXTRA EXPERIMENTS

Table 7: Test accuracy of text classification tasks under group-wise clipping styles.

| Model | Method | | $\epsilon = 3$ | | | | $\epsilon = 8$ | | | |
|---|---|---|---|---|---|---|---|---|---|---|
| | | | MNLI | QQP | QNLI | SST2 | MNLI | QQP | QNLI | SST2 |
| RoBERTa -base | RGP | Yu et al. (2021b) | -/- | - | - | - | 80.5/- | 85.5 | 87.2 | 91.6 |
| | all-layer ($M = 1$) | (Li et al., 2021) | 82.45/82.99 | 85.56 | 87.42 | 91.86 | 83.20/83.46 | 86.08 | 87.94 | 92.09 |
| | all-layer ($M = 1$) | (Bu et al., 2022b) | 83.22/83.21 | 85.76 | 86.91 | 92.32 | 83.82/83.55 | 86.58 | 87.85 | 92.43 |
| | block-wise ($M = 12$) | ours | 82.55/83.19 | 84.14 | 85.94 | 91.74 | 83.06/83.29 | 84.73 | 86.40 | 91.97 |
| | layer-wise ($M = 103$) | ours | 82.02/82.56 | 83.26 | 85.85 | 91.40 | 82.24/82.84 | 83.49 | 86.42 | 92.09 |
| | layer-wise* ($M = 103$) | He et al. (2022) | 82.83/83.27 | 85.67 | 86.13 | 92.03 | 83.70/83.97 | 86.23 | 87.13 | 92.40 |
| | param-wise ($M = 203$) | ours | 81.63/82.10 | 82.52 | 85.25 | 91.28 | 82.22/82.49 | 82.80 | 86.09 | 91.63 |
| RoBERTa -large | RGP | Yu et al. (2021b) | -/- | - | - | - | 86.1/- | 86.7 | 90.0 | 93.0 |
| | all-layer ($M = 1$) | Li et al. (2021) | 86.43/86.46 | 86.43 | 90.76 | 93.04 | 87.02/87.26 | 87.47 | 91.10 | 93.81 |
| | all-layer ($M = 1$) | (Bu et al., 2022b) | 86.27/86.67 | 86.7 | 91.01 | 93.92 | 87.07/87.16 | 87.47 | 91.45 | 94.61 |
| | block-wise ($M = 24$) | ours | 87.26/87.28 | 85.81 | 89.86 | 94.15 | 87.54/87.29 | 86.55 | 90.78 | 94.61 |
| | layer-wise ($M = 199$) | ours | 86.37/86.66 | 84.78 | 89.60 | 94.38 | 86.53/86.93 | 85.22 | 90.10 | 94.50 |
| | layer-wise* ($M = 199$) | He et al. (2022) | 87.10/87.20 | 86.80 | 89.80 | 93.87 | 87.67/87.57 | 87.20 | 90.77 | 94.03 |
| | param-wise ($M = 395$) | ours | 86.47/86.38 | 84.49 | 89.11 | 93.72 | 86.43/86.39 | 85.17 | 89.84 | 94.27 |

