# OpenReview forum: "On the efficacy of group-wise clipping in differentially private optimization"
_ICLR.cc/2024/Conference — ICLR 2024 Conference Withdrawn Submission_

### Official Review · Reviewer_fLMf · 2023-11-02

**Soundness:** 2 fair
**Presentation:** 3 good
**Contribution:** 2 fair
**Rating:** 3
**Confidence:** 4

**Summary:**

The paper discusses recent advancements in differentially private (DP) deep learning, focusing on large vision and language models with millions to billions of parameters. The authors find that different group-wise clipping styles offer an accuracy-memory trade-off. While all-layer clipping is commonly used and provides better accuracy, it requires more memory compared to group-wise clipping. The paper formalizes this trade-off through convergence theory and complexity analysis. Importantly, it demonstrates that the accuracy gap between group-wise and all-layer clipping decreases with larger models, while the memory advantage of group-wise clipping remains, allowing DP optimization of large models with high accuracy and low peak memory usage.

**Strengths:**

The paper addresses an important aspect of DP deep learning, namely gradient clipping, which is crucial for privacy-preserving training of large models. It thoroughly explored the design space of group-wise clipping styles for various learning tasks.

Empirical Experiments: The paper includes a good set of experiments to support its claims.

**Weaknesses:**

**Motivation of Group-Wise Clipping**: In the abstract, the paper claims The paper lacks a clear and strong motivation for why group-wise clipping is a necessary or valuable alternative to all-layer clipping as **all group-wise clipping enjoy almost the same training speed as the standard non-DP optimization**. Meanwhile the memory cost does not differentiate too much across various grouping choices, either (see Table 3 and Figure 5).

**Confusing measures**: There are several terms used across the paper, e.g., time complexity, training speed, memory cost. The paper should define them clearly whether they are theoretically or empirically computed. If empirically, the training speed and the memory cost are jointly affected by the setup of the batch size, model size and the model architecture. Book-keeping technique would store the backward gradients on the output of each operation, the same as storing the activations, which may have memory problem when the batch size is large.

As a following weak point, the paper does not talk about the implementation detail and wall-clock training speed comparison.  This is because the non-uniform grouping is complex to implement and the wall-clock training speed is the ultimate measure for different choices.
The cost of searching the best non-uniform grouping is not counted.

**Relevance of Theory**: The theoretical analysis may not provide sufficient insights into practical scenarios. The upper bound gets sub-linearly (sqrt) worse as the number of the groups increases, which is not reflected in real experiments. Theorem 2 is a bit trivial and does not convey much information related with the target of the paper.


**Experiments presentation**: The experiments are cherry picked in the main text. It seems that the results of the paper are not as good as the result of He et al. 2022 in Appendix C, which are excluded from the main text. Moreover, all the experiments consider the fine-tuning setting, which is not clearly stated in the main text. There lack training scratch experiments for full comparison.

**Questions:**

Questions about the experiment results.  In Table 3, the memory cost increases as you increases the number of groups for QNLI RoBERTa-base. This contradicts with theory analysis and all other experiments. Can the authors explain why this happens?

**Details Of Ethics Concerns:**

The paper studies how to protect the privacy of dataset when training with large models. It explores new algorithms/methods for the privacy protection.

---

> ### Author Response · Authors · 2023-11-20
>
> We thank the reviewer's comment but there are lot of mis-understandings and false statements in it, which unfairly affect the scoring, especially point 6.
>
> 1. The paper lacks a clear and strong motivation for why group-wise clipping is a necessary or valuable alternative to all-layer clipping...Meanwhile the memory cost does not differentiate too much across various grouping choices
>
> Response: It is important to understand the pros and cons of any group-wise clipping, whereas group-wise clipping is over-looked. E.g. layer-wise clipping can be 10-20% more memory efficient than all-layer clipping: **imagine saving more than a month of computation out of a 12-month project**. Also, Remark 3.1 states the unique advantage of applying layer-wise clipping to DP distributed learning, when inter-layer communication is expensive and complicated.
>
> 2. There are several terms used across the paper, e.g., time complexity, training speed, memory cost. The paper should define them clearly whether they are theoretically or empirically computed.
>
> Response: We will revise it to clarify the mis-understanding. The time/space complexity are theoretical values, whereas the training speed and memory cost are empirical. The theoretical values are used for comparison, i.e. algorithm A is *faster in theory* than algorithm B if the time complexity of A is lower. But the same algorithm implemented differently on different devices could result in different training speed.
>
> 3. Book-keeping technique would store the backward gradients on the output of each operation, the same as storing the activations, which may have memory problem when the batch size is large.
>
> Response: **We are not sure why this sentence is a weakness?** Maybe the reviewer is missing the part that different groupings have different definition of "each operation". For instance, layer-wise clipping treats one layer as one operation but all-layer clipping treats all layers as one operation (visualized in Figure 2). This difference leads to the memory difference.
>
> 4. As a following weak point, the paper does not talk about the implementation detail and wall-clock training speed comparison.
>
> Response: **This is not true**. We have explicitly provided the implementation details in Algorithm 1 and the wall-clock training time in Figure 5. We will release the code upon acceptance but certainly cannot violate the anonymity before then.
>
> 5. The theoretical analysis may not provide sufficient insights into practical scenarios. The upper bound gets sub-linearly (sqrt) worse as the number of the groups increases, which is not reflected in real experiments.
>
> Response: **It is false to say "The upper bound gets sub-linearly (sqrt) worse as the number of the groups increases"**. It depends. In Theorem 1, when $\xi>>M$, the upper bound gets $M^{1/4}$ worse as the number of the groups increases. This is not contradicted by real experiments: changing from all-layer clipping $M=1$ to layer-wise clipping (say $M=81$) only slows down the convergence by 3 times, *if the upper bound is tight*. We note that our upper bound is the first of its kind and may be improved in terms of tightness.
>
> 6. The experiments are cherry picked in the main text. It seems that the results of the paper are not as good as the result of He et al. 2022 in Appendix C, which are excluded from the main text.
>
> Response: **This is an unfair accusation because the referred paper is not reproducible at all!** We did not cherry pick the experiments and it is not our fault to not compare to their results. The referred appendix are working with a non-public model (GPT-3, which is proprietary at this moment), open-sourcing 0% of code (no Github repo) and giving no implementation details (e.g. context length, pre-processing details).
>
> 7. Moreover, all the experiments consider the fine-tuning setting, which is not clearly stated in the main text. There lack training scratch experiments for full comparison.
>
> Response: **This comment is partially irrelevant.** One of the contributions of this paper is the time/space complexity (or speed and memory) analysis and experiments. These have nothing to do with the initialization, i.e. regardless we are pre-training or fine-tuning. In addition, our convergence analysis also does not distinguish between pre-training and fine-tuning. Therefore, we don't believe it is necessary to constrain ourselves in the fine-tuning setting, and it should be sufficient to declare that we are fine-tuning (as in Appendix B) only for those experiments where there is a difference.
>
> 8. Questions about the experiment results. In Table 3, the memory cost increases as you increases the number of groups for QNLI RoBERTa-base. This contradicts with theory analysis and all other experiments. Can the authors explain why this happens?
>
> Response: **We did not increase the number of groups at all.** It is always two-group clipping in Table 3 (see the caption).

---

### Official Review · Reviewer_yntr · 2023-11-04

**Soundness:** 2 fair
**Presentation:** 2 fair
**Contribution:** 2 fair
**Rating:** 5
**Confidence:** 5

**Summary:**

This paper studies group-wise clipping for optimization under differential privacy.

**Strengths:**

The issues discussed in this article regarding optimization under DP are timely and critical. The performance loss caused by DP necessitates urgent solutions for these problems.

**Weaknesses:**

The paper lacks novelty as the proposed clipping method is an extension of the existing Book-Keeping
technique Bu et al. (2022c). Furthermore, the convergence analysis relies on smoothness assumptions.

I also disagree with the authors' perspective that "Differentially private (DP) optimization of deep learning models has enjoyed amazing accuracy and rigorous guarantees against privacy risks." From my knowledge, accuracy loss remains a significant obstacle, which is also the problem this paper aims to address.

**Questions:**

Are there any hyperparameters that need to be tuned for the proposed clipping methods? If so, do these adjustments come at an additional privacy cost? Has the paper reported these associated costs?

---

> ### Author Response · Authors · 2023-11-20
>
> We thank the reviewer's comment. Here is a point-to-point response.
>
> 1. The paper lacks novelty as the proposed clipping method is an extension of the existing Book-Keeping technique Bu et al. (2022c). Furthermore, the convergence analysis relies on smoothness assumptions.
>
> Response: We agree that the proposed method is an extension of existing BK technique. However, we disagree that the paper lacks novelty because our clipping is not our only contribution. We provide the first layer-wise clipping convergence analysis in non-convex (though smooth) setting, which indeed helps our understanding in the non-smooth setting, as shown in all empirical experiments. The system design from accuracy-speed-memory perspectives is new.
>
> 2. I also disagree with the authors' perspective that "Differentially private (DP) optimization of deep learning models has enjoyed amazing accuracy and rigorous guarantees against privacy risks." From my knowledge, accuracy loss remains a significant obstacle, which is also the problem this paper aims to address.
>
> Response: To be more rigorous, we would state "Differentially private (DP) optimization of deep learning models has enjoyed amazing accuracy and rigorous guarantees against privacy risks in a range of tasks." The accuracy loss has been significantly mitigated in the DP fine-tuning scenarios.
>
> 3. Are there any hyperparameters that need to be tuned for the proposed clipping methods? If so, do these adjustments come at an additional privacy cost? Has the paper reported these associated costs?
>
> Response: There is no hyperparameter to tune, as indicated in Equation (4).
>
> We are happy to extend the discussion and hope the reviewer can consider raising the score if satisfied.

---

### Official Review · Reviewer_nK9w · 2023-11-05

**Soundness:** 2 fair
**Presentation:** 2 fair
**Contribution:** 2 fair
**Rating:** 5
**Confidence:** 3

**Summary:**

This paper studies the group-wise clipping approach in DP, and gives analysis on its convergence and its algorithmic relation to back-propagation. The authors also analyze the system wise metrics such as peak memory profile usage. Empirical results are given on GPT2 and ViT models.

**Strengths:**

* The paper provides detailed analysis to the group-wise clipping technique in DP domain, some of the conclusions are interesting to this field.
* The authors give both insights from theory and system perspectives.
* The authors also set up new baseline results, which could potentially be a good reference for further work in this space.

**Weaknesses:**

* From the peak memory profile results, i.e. Table 3 and Figure 5, it looks like the peak memory usages for different boundaries are pretty close (in general less than 2 GB). I'm not sure how much this can lead to faster training and larger batch sizes. For example, what is the new batch size that can be used, and how much speed up we gain? Some real-world numbers here could be beneficial.
* From Theorem 1, it looks like the AUTO algorithm obtains the same convergence speed compared to the standard SGD. However, the standard SGD does not require per-sample gradient to be symmetric about the oracle gradient as shown in Assumption 4.3. I wonder if this is critical for AUTO to get on-par convergence speed to SGD? What will the convergence rate be like without such assumption?
* In the paper, the authors object to the conclusion of https://arxiv.org/pdf/2212.01539.pdf with a self-designed group-wise clipping algorithm for faster training speed. However, I don't see too much evidence supporting this. Could you show a convergence curve?

**Questions:**

Please refer to the weaknesses section.

---

> ### Author Response · Authors · 2023-11-20
>
> We thank the reviewer's comment. Here is a point-to-point response.
>
> 1. I'm not sure how much this can lead to faster training and larger batch sizes. For example, what is the new batch size that can be used, and how much speed up we gain? Some real-world numbers here could be beneficial.
>
> Response: In general we expect to see 10-20% speed up due to larger batch size towards finer-grained grouping, e.g. layer-wise clipping. We provided these numbers in Table 3 (e.g. RoBERTa is more severely affected by the choice of group-wise clipping) and Figure 3. We note that a saving of 10-20% is significant in large model training, which particularly favors layer-wise clipping (see Remark 3.1).
>
> 2. it looks like the AUTO algorithm obtains the same convergence speed compared to the standard SGD. However, the standard SGD does not require per-sample gradient to be symmetric about the oracle gradient as shown in Assumption 4.3. I wonder if this is critical for AUTO to get on-par convergence speed to SGD? What will the convergence rate be like without such assumption?
>
> Response: We highlight that DP algorithms converge at the same *asymptotic* speed up to a constant in $O$, hence the standard SGD can still be significantly faster if SGD's loss is like 1/t but DP-SGD's is like 100/t. In the current literature, due to the complicated form of per-sample gradient clipping, it is challenging if not impossible to analyze the convergence without such assumption.
>
> 3. In the paper, the authors object to the conclusion of https://arxiv.org/pdf/2212.01539.pdf with a self-designed group-wise clipping algorithm for faster training speed. However, I don't see too much evidence supporting this. Could you show a convergence curve?
>
> Response: While there are many conclusions in that paper, we specifically object to that the layer-wise clipping is faster than all-layer clipping; we claim any group-wise clipping has the same speed per iteration. In words, any group-wise (not only our own-designed) takes the same number of hours to optimizer for 10000 iterations. Regarding the convergence curve in our Theorem 1, this is different from wall-clock time, i.e. this is the loss v.s. number of iterations, which is missing from that paper at all. We are happy to extend the discussion on either type of speed.
>
> We hope the reviewer are happy with our response and consider raising the score.

---

### Official Review · Reviewer_RBj1 · 2023-11-06

**Soundness:** 3 good
**Presentation:** 2 fair
**Contribution:** 3 good
**Rating:** 6
**Confidence:** 3

**Summary:**

Recent advances in differentially private deep learning have improved accuracy, memory efficiency, and training speed for large models. This paper focuses on per-sample gradient clipping methods in DP optimization. It finds that different clipping styles have similar time complexity but trade off accuracy and memory usage. All-layer clipping offers better accuracy but requires more memory than group-wise clipping. As models grow larger, the accuracy gap narrows, while the memory advantage of group-wise clipping remains, making it suitable for efficient DP optimization of large models.

**Strengths:**

+ It's an interesting paper that leverages memory-accuracy tradeoff of group-wise dp optimization with different granularity.
+ The key observation about dS doesn't depend on dW so that the computational time doesn't depend on m provides great ml-sys type of insights.
+ The ViT experiments on Cifar100 is convincing.

**Weaknesses:**

- The presentation needs some work. The paper contains multiple contributions and a lot prior work / settings, which was clear in the introduction, but very confusing in later sections. For example, I was very confused about the equal time efficiency part because authors wrote this contribution directly so I thought that was the previous design. Specifically, if this is the contribution, I would sign-post it at the beginning of section 3 what are the conventional wisdom and why a simple analysis on computational dependency graph (you don't need dW to derive dS) would do the work. It requires many passes of reading and reasoning to get the point.
 - The presentation of experiment section is poor. Also ImageNet is mentioned at the beginning but the experiments don't have it? In addition, cifar10/100 (better imagenet) are convincing Image baselines, but why using E2E dataset in the last experiment 1) it is not popular for decoder only model 2) you didn't benchmark the peak memory for gpt. Also I understand you benchmarked peak memory before, but table 5 and 6 better have acc and peak mem side by side.

**Questions:**

I'm curious in authors' view, is this 1-2 GB memory difference significant? Or in another word, is this an important tradeoff worth studying to begin with?

---

> ### Author Response · Authors · 2023-11-20
>
> We thank the reviewer's comment. Here is a point-to-point response.
>
> 1. For example, I was very confused about the equal time efficiency part because authors wrote this contribution directly so I thought that was the previous design. Specifically, if this is the contribution, I would sign-post it at the beginning of section 3 what are the conventional wisdom and why a simple analysis on computational dependency graph (you don't need dW to derive dS) would do the work.
>
> Response: Yes it is our new contribution, not previous design, to make DP algorithms with different group-wise clippings equally fast. This is achieved by extending on Book-Keeping algorithm that only works for all-layer clipping. We are happy to make the change in the next revision.
>
> 2. ImageNet is mentioned at the beginning but the experiments don't have it? ..... why using E2E dataset in the last experiment 1) it is not popular for decoder only model 2) you didn't benchmark the peak memory for gpt.
>
> Response: ImageNet is not experimented due to lack of pre-training datasets; note that the benchmarks are pre-trained on Google's proprietary dataset JFT, which is non-public and non-reproducible. We used E2E dataset as a popular DP-GPT benchmark in "Large Language Models Can Be Strong Differentially Private Learners", "Automatic Clipping: Differentially Private Deep Learning Made Easier and Stronger", "Differentially Private Fine-tuning of Language Models", etc. We would like to add the benchmark of peak memory for GPT in the next revision, which doesn't fit in the current tables.
>
> 3. I'm curious in authors' view, is this 1-2 GB memory difference significant? Or in another word, is this an important tradeoff worth studying to begin with?
>
> Response: We thank the reviewer for this question. To put this into perspective, we are looking at 2-4 GB memory difference (by Table 3) out of 10-20 GB total memory usage, meaning 10-20% saving that is transferrable to at least 10% training speedup. This is significant since large model training is expensive: saving 10% means millions out of tens of millions of dollars, or 1 month from a year-long project.
>
> We also highlight that the memory analysis is primarily related to 1 out of our 5 contributions (the 4-th), which also introduce a new convergence theory, time efficiency and the compatibility of DP algorithms to distributed learning.